# Modelling Polarization Effects in a CdZnTe Sensor at Low Bias

**DOI:** 10.3390/s23125681

**Published:** 2023-06-17

**Authors:** Jindřich Pipek, Roman Grill, Marián Betušiak, Kris Iniewski

**Affiliations:** 1Institute of Physics, Faculty of Mathematics and Physics, Charles University, Ke Karlovu 5, CZ-121 16 Prague, Czech Republic; 2Redlen Technologies Inc., 1763 Sean Heights, Saanichton, BC V8M 0A5, Canada

**Keywords:** CdZnTe, radiation detector, high flux, polarization

## Abstract

Semi-insulating CdTe and CdZnTe crystals fabricated into pixelated sensors and integrated into radiation detection modules have demonstrated a remarkable ability to operate under rapidly changing X-ray irradiation environments. Such challenging conditions are required by all photon-counting-based applications, including medical computed tomography (CT), airport scanners, and non-destructive testing (NDT). Although, maximum flux rates and operating conditions differ in each case. In this paper, we investigated the possibility of using the detector under high-flux X-ray irradiation with a low electric field satisfactory for maintaining good counting operation. We numerically simulated electric field profiles visualized via Pockels effect measurement in a detector affected by high-flux polarization. Solving coupled drift–diffusion and Poisson’s equations, we defined the defect model, consistently depicting polarization. Subsequently, we simulated the charge transport and evaluated the collected charge, including the construction of an X-ray spectrum on a commercial 2-mm-thick pixelated CdZnTe detector with 330 µm pixel pitch used in spectral CT applications. We analyzed the effect of allied electronics on the quality of the spectrum and suggested setup optimization to improve the shape of the spectrum.

## 1. Introduction

Nowadays, CdTe and CdZnTe are the most advanced room-temperature semiconducting materials suitable for X-ray detection. CdZnTe is being used in X-ray computed tomography [1], gamma-ray nuclear medicine [2], baggage scanning [3], and non-destructive testing [4] applications. One of the most important requirements for radiation sensors used in high-flux photon counting applications is their ability to operate in an intense and rapidly changing X-ray environment. These sensors need to sufficiently sustain high fluxes of incoming X-rays of the order of 600 Mcps/mm^2^ while maintaining a short enough charge collection time and high temporal stability [5]. Due to the low mobility of holes compared to that of electrons and substantial hole trapping, traditionally, the use of CdTe and CdZnTe materials under these intense irradiation environments has been limited [6,7]. Suffering from severe X-ray dynamic polarization because of electric field distortion and collapse, historical high-Z sensors were unable to operate under relatively low X-ray flux levels. In recent years, however, these issues have been studied extensively, and significant gains in sensor performance have been achieved [8,9,10,11,12]. These advancements were mainly due to improvements in crystal growth (by suppressing the density of the deep hole traps) and fabrication quality (by minimizing the fabrication-induced surface states). In particular, important progress in the charge transport properties and uniformity has been made. High electron mobility–lifetime products of electrons ${µ}_{e}\tau_{e}$ greater than $10^{-2}cm^{2}/V$ were achieved [13]. Simulations of semiconductor radiation detectors were studied in [14,15,16,17,18] to obtain charge transport properties. Charge transport in pixelated detectors was studied in [19,20,21], where the effects of charge sharing and material quality on detector performance were investigated.

High-flux sensors typically require an operating bias U of the order of 1000 V. Assuming that the typical value for the mobility of the electrons is $µ=1000\;cm^{2}/Vs$, then for a high voltage bias of U=1000 V, the electric field for a L=2 mm thick detector is ϵ=5 kV/cm, which corresponds to the electron transit time through the whole detector thickness of 40 ns. This high electric field might cause some long-term reliability concerns; therefore, it would be desirable to reduce its value. In addition, the requirement of generating high-voltage U causes limitations for portable scanning equipment. It is, therefore, important to study the minimum electric field required for proper sensor operation. The main problems arising due to a reduced electric field are the slowing down of the drift velocity and the extension of the transit time. Reducing the electric field leads to a decrease in the amount of collected charge and the count numbers obtained.

In this paper, we simulate the charge transport inside the sensor, evaluate the properties of carrier traps and demonstrate the effects caused by a reduced electric field. We define the critical bias $U_{c}$ as producing sufficient collection efficiency to operate under high X-ray flux and characterize the sensing quality of the sensor operating under such conditions. The simulation fits the commercial detector to obtain additional insights into the physical effects on the detector under low electric fields. The existing literature does not cover this topic.

## 2. Experiment

Several high-flux sensors with various pixel pitches and 2 mm thickness were fabricated and experimentally analyzed using a variety of characterization techniques. Here, a semi-insulating commercial-grade CdZnTe detector with the dimensions of $10\times 10\times 2\;{mm}^{3}$ with 330 µm pitch used in spectral Computed Tomography high-flux applications is studied. The CdZnTe sensor used in this study reaches the highest count rate performance known in the X-ray detection of High-Z materials, achieving rates over 1000 Mcps/mm^2^ [22]. The sensor has been extensively characterized using PICTS (Photo-Induced Current Transient Spectroscopy, Corema resistivity measurements, DLTS (Deep Level Transient Spectroscopy), and the Pockels technique. We have found that the Pockels effect, which builds on the linear electro-optic effect where the refractive index of a medium is modified in proportion to the applied electric field strength, was particularly useful in measuring the electric field under various high-voltage conditions.

The detector has two opposite electrodes: the cathode is planar, and the anode is pixelated [23]. The detector is characterized using the Pockels technique [24,25,26], from which the electric field profile is obtained for several applied biases in the dark and under different X-ray fluxes. A standard X-ray tube set-up with a kVp of 120 kVp is used [27]. The respective X-ray spectrum and attenuation coefficient of CdZnTe are shown in Figure 1 [28,29]. The scheme of the detector with incoming X-rays from the cathode side is shown in Figure 2, where the typical electric field profile of a polarized detector is outlined.

The attenuation coefficient of X-ray α(E) is fitted to the interval of 10 keV–120 keV by a fitting function, as follows:(1)$$\alpha \left(E\right)=[808\times 10^{2.65}+114.6{\left(E48K\right)}^{2.65}\Theta \left(E-E48K\right)+\;+94.7{\left(E52K\right)}^{2.65}\Theta \left(E-E52K\right)]/E^{2.65}$$
where Θ(*E*) is the Heaviside step function. The fitted data, including energies *E*48*K* = 26.7112 keV and *E*52*K* = 31.8138 keV, are taken from [28].

Detector performance was evaluated using a photon-counting ASIC that is used in the detector module for Spectral Computed Tomography. The ASIC has 864 (24 × 36) identical channels. Each channel has a typical charge sensitive amplifier (CSA) with a feedback capacitor. The output of the CSA is amplified by the shape amplifier and compared in the comparator to the set threshold(s). When the signal crosses pre-defined threshold levels (for example, 16 keV), the clock starts, and the circuits samples for the sampling time $t_{s}=16\;ns$ of the collected charge. If the charge is saturated in this period, the correct value is obtained. If saturation is not reached, the circuit reads a lower value, creating an energy error. This feature is called a ballistic deficit.

## 3. Simulation of the Detector Performance at Low Bias

Detector polarization induced by hole trapping under high-flux X-ray excitation leads to a significant reduction in the electric field near the pixelated anode. An extended charge collection time amplifies the ballistic deficit and limits the maximum accessible X-ray flux that can be distinguished in spectrally sensitive electronics. An extended transit time also constrains high-flux applications, leading to pileups and the loss of energy resolution. In our case, the transit time is still short enough to minimize the charge sharing between the pixels due to diffusion broadening. Considering that the transit time $t_{r}$ is shorter than 200 ns for a bias of >200 V, we estimate the diffusion broadening $∆x\approx \sqrt{D_{e}t_{r}}<20\;\mu m$, which is much less than the pixel pitch of 330 μm. Thus, we do not add the correction to the charge sharing here, i.e., the signal of only one pixel is considered. In this paper, we model space charge effects due to high-flux X-ray irradiation with the spectrum plotted in Figure 1.

In the charge transport model used for the charge collection calculation, we only consider the drift current of electrons and describe the current waveform *I(t)* in one pixel in a pixelated detector with the electric field profile with the following formula:(2)$$I\left(t\right)=Q\left(t\right)\mu_{e}\epsilon \left(x\left(t\right)\right)E_{W}\left(x\left(t\right)\right),$$
where $\mu_{e}$ is electron drift mobility, ϵ(x) is the electric field at the depth x, and $E_{W}(x)$ is the weighting field calculated for the pixelated detector according to [30]. Drifting photogenerated charge $Q(t)=Q_{0}exp(-t/\tau_{e})$ is attenuated with the electron lifetime $\tau_{e}$ that is taken to be constant, i.e., the space charge is considered to be independent, for simplicity. The weighting field against the middle of the pixel is considered. The electric field is obtained through the fitting of the Pockels effect data.

Starting with Equation (2), the numerical treatment is performed according to the following steps. 

The position of the drifting charge excited at the cathode x(t) is calculated by integrating the following kinetic equation:(3)$$\int_{0}^{x(t)}\frac{dx\prime }{\epsilon \left(x\prime \right)}dx\prime =\mu_{e}t.$$Normalized charge $\tilde{q_{0}}\left(t\right)$ excited at the cathode and collected in time t, considering the attenuation of the collected charge due to the RC time $\tau_{RC}$ is calculated by integrating the following equation:(4)$$\frac{d\tilde{q_{0}}}{dt}=-\frac{\tilde{q_{0}}}{\tau_{RC}}+\frac{I(t)}{Q_{0}}.$$The RC time is usually much longer than the transit time, and the first term on the right-hand side of Equation (4) may be neglected.Utilizing the linearity of Equation (4), we may conveniently generalize $\tilde{q_{0}}\left(t\right)$ to the case of the charge collection of the normalized charge excited in the detector’s interior that may be expressed as follows:(5)$$\tilde{q}\left(t,t_{d}\right)=\left[\tilde{q_{0}}\left(t\right)-\tilde{q_{0}}\left(t_{d}\right)e^{-\frac{t-t_{d}}{\tau_{RC}}}\right]e^{\frac{t_{d}}{\tau_{e}}}\Theta \left(t-t_{d}\right).$$$t_{d}$ is the drift time representing the drift delay of the charge excited at the cathode to the depth d in the detector where X-ray photon absorption has occurred. $t_{d}$ is linked to d by Equation (3). The scaling of the drift of the charge through the detector by the drift time appeared to be useful in the calculations. This achievement enables us to significantly simplify the enumeration when the calculation of the collected charge following the excitation wherever in the bulk may be derived from the collected charge excited near the cathode without the additional solution of differential Equation (4).We simulate the processing of the collected charge by the electronic circuit. This circuit is characterized by the threshold energy $E_{t}=16\;keV$ at which the charge sampling starts, and the sampling time $t_{s}$, defining the time window of the charge collection. The process is simulated in two steps. At first, the collected charge expressed as $Q_{0}\tilde{q}\left(t,t_{d}\right)$ is monitored, waiting for the time at which it exceeds the energy threshold. Subsequently, the collected charge $q_{c}\left(t_{d}\right)=Q_{0}\tilde{q}\left(t+t_{s},t_{d}\right)$ is evaluated.Having $q_{c}\left(t_{d}\right)$, we may start with the construction of the spectra. The normalized spectrum $S_{m}(ch,E_{X})$, indexed by a channel number ch and excited with a monochromatic X-ray photon with energy $E_{X}$, is obtained by the sum of the contributions to the spectrum excited by the photon absorbed in a specific depth of the detector. We proceed with the following loop in i
(6)$$S_{m}^{\left(i\right)}\left({ch}_{i},E_{X}\right)=S_{m}^{\left(i-1\right)}\left({ch}_{i},E_{X}\right)+\alpha \left(E_{X}\right)e^{-\alpha \left(E_{X}\right)x\left(t_{di}\right)}\frac{∆x_{i}}{∆t}.$$
where the channel number ${ch}_{i}$ is defined by the collected charge
(7)$${ch}_{i}=\left\lfloor \frac{q_{c}\left(t_{di}\right)}{∆Q}+\frac{1}{2}\right\rfloor ,$$
where lower brackets represent the floor function that returns the integer part to yield ${ch}_{i}$. The index i scales the drift time $t_{di}$ through the detector thickness, and ΔQ defines the width of one channel and $\Delta x_{i}=x\left(t_{di}\right)-x\left(t_{d(i-1)}\right)$. In real calculations, to obtain smooth curves, we divide the energy axis into many more channels than used in the experiments, i.e., ΔQ is sufficiently small. The channel is assigned to the apparent photon energy
*E* = *ch* Δ*Q*
*E_X_*/*Q*_0_ = *ch* Δ*E_X_*.(8)
which can be simplified by defining the channel energy width $\Delta E_{X}=\Delta Q\;E_{X}/Q_{0}$. This option is further used in calculations, allowing us to join photoexcited charge with energy units.The last step of the treatment is the construction of the full X-ray spectrum S(E). It is calculated by integrating the contributions of photons at a specific energy weighted with the corresponding radiation intensity. The final formula reads as follows:(9)$$S\left(E\right)=\int S_{m}\left(ch,E_{X}\right)I_{0}\left(E_{X}\right)dE_{X},$$
where E and ch are interconnected by Equation (8).

## 4. Results and Discussion

### Discussion

To simulate the charge transport and the space charge formation inside the detector, we numerically solved the drift–diffusion equation coupled with Poisson’s equation [31]. This allowed us to obtain the profiles of space charge, electric field, and energy level occupancies. Since the polarization phenomena are caused, in our case, by high flux irradiation, we disregarded electric-contacts-induced polarization [9,12,26,31]. We used Ohmic contacts that cause zero band bending under contacts in simulation to prevent detector charging in the dark. Possible deviation of the contacts from an Ohmic character does not affect the model unless the injection current approaches the value of the photocurrent. Obviously, the use of a strongly injecting contact would be improper because an enhanced injected current would induce an enormous noise, making the detector useless. The simulations were performed on a sensor with a thickness of L=2 mm. Electron mobility $\mu_{e}=1000\;{cm}^{2}/Vs$ and hole mobility $\mu_{h}=80\;cm^{2}/Vs$ are consistent with common values measured in CdZnTe [32].

Typical electric field profiles measured using Pockels effect in the dark (no X-ray) and 16 Mcps/mm^2^ and 80 Mcps/mm^2^ high-flux X-ray irradiation are shown in Figure 3a; the simulated electric field profiles are shown in Figure 3b. As expected, the electric field is uniform under dark conditions, but it degrades near the anode under high-flux excitation. The simulated electric field profiles show good agreement with the experiment, validating the used defect model (Figure 3b).

X-ray excitation intensity representing the rate of e h pairs generation throughout the sample is calculated according to the following formula:(10)$$I_{exc}(x)=\frac{1}{E_{b}}\int I_{xray}(E)\alpha \left(E\right)E\exp(-\alpha \left(E\right)x)dE,$$
where $I_{xray}$ is X-ray photon flux density, E is the energy of the X-ray photon, and $E_{b}=$ 4.5 eV is the average energy of the formation of e h pairs. The respective excitation intensity $I_{exc}$ plotted in Figure 4 shows dominant excitation under the cathode and a fast decrease toward the anode.

The defect model fitting the electric field profile consists of one electron and one hole deep trap whose parameters are defined in the defect scheme in Figure 5. The Fermi level was fixed in midgap $E_{F}=E_{C}-0.775$ eV. Since we performed only steady-state experiments at a constant temperature, capture cross-sections cannot be determined. As discussed in [33], steady-state experiments do not offer a way to determine the capture cross-section of the traps.

The corresponding space charge density is shown in Figure 6. While the hole trapping in the hole trap causes the positive space charge and the tilt of the electric field throughout the sample, the presence of the electron trap causes a gradual decrease in the positive space charge along the sample thickness and even negative space charge formation near the anode.

Subsequent measurements were performed using a standard X-ray tube with a 120 kVp setup at a photon flux of 20 Mcps/mm^2^.

The counts vs. bias were measured at a count rate of 20 Mcps/mm^2^, as shown for one representative pixel in Figure 7. The counts for $U<U_{c}$ are approximated using a straight line, and the point at which maximum counts were achieved is considered to be $U_{c}\approx 265\;V$. This procedure was repeated for each pixel, and the distribution of $U_{c}$ is shown in Figure 8. The decrease in measured counts at low bias is not mainly caused by carrier trapping but by the ballistic deficit of low energy X-ray photons, which do not cross the threshold level, frequent pileups, and charge sharing between pixels [19], which is accentuated at the low electric field near the pixelated anode.

In Figure 9, we show the current waveforms calculated using Equation (2) with the electric field obtained from the numerical simulation with the X-ray photon absorbed close to the cathode, and the use of the weighting field is plotted in Figure 6. As is common in pixelated detectors, the weighting field is low near the cathode and rapidly increases near the anode due to specific pixel sensitivity. This feature allows us to eliminate holes from the calculations since only a small number of holes is generated near the anode in the area with the large value of the weighing field. Simultaneously, the low mobility of holes yields a low contribution to the charge collection at the chosen short sampling time.

The current waveforms in Figure 9 were simulated with near-cathode absorption, this does not affect the result since the weighting field is small there, and only electrons that drift near the anode contribute to the signal. This is demonstrated in Figure 10, where the collected charges for photons absorbed near the cathode (red) and deep inside the detector 1.44 mm (blue) are shown. These curves are synchronized in time so that the arrival of charge to the anode occurs at the same time, 146 ns, regardless of the depth of the charge creation. The lower detected counts in Figure 7 at low bias are due to the long transit time compared to the sampling time. Carrier losses due to the limited lifetime have a minor effect in this case since the drift time remains much shorter than the lifetime, even at the considered low bias. In the case of zero space charge, the shoulder at the count number in Figure 7 would be shifted significantly to the lower bias, and the segment of the respective curve would be flat in the plotted region.

Based on the critical bias shown in Figure 7, we have chosen a bias of 300 V as the characteristic value of low bias considered for the utilization of the detector for X-ray spectroscopy. We calculated the X-ray spectrum detected by the detector simulated by the theory presented in items 1–6 in Section 3, assuming that the detector polarization is due to the high-flux X-rays at 20 Mcps/mm^2^ and common parameters characterizing the circuit $\tau_{RC}=1\;µs$, which, in our case, is much longer than the sampling time of $t_{s}=16\;ns$, and its effect is negligible. The corresponding spectrum, together with the original spectrum, is plotted in Figure 11. We may identify two distinct regions in the spectrum. While the low-energy part fits the original spectrum very well, the high-energy counts are collected at nearly the same energy of about 45 keV, which manifests as the large peak in that region. The effect is induced by the ballistic deficit, which is caused by the slow charge collection, which is not completed during the 16-ns sampling time.

The details of the charge collection are demonstrated in Figure 12, where the collections of the charge induced by the photons with three different energies are depicted. The red curve represents the relative collected charge whose shape remains the same at all photon energies. The charge collection by the electronics is critically affected by the relative position of the threshold marked with horizontal lines representing a threshold of 16 keV. The first case (A) represents the photon with an energy smaller than the threshold of 16 keV. Since the collected charge curve is lower than the threshold, no sampling event occurs, and the charge is not recognized by the electronics. Inasmuch as the used X-ray spectrum shows very low intensity below 16 keV, this defect does not manifest in the calculated spectrum shown in Figure 11. Case (B) represents the photon with energy larger than the threshold. As soon as the collected charge is greater than the threshold, the trigger starts and samples for the sampling time $t_{s}$ (blue bar), after which the charge is collected (marked with x). Since the collection terminates after the transit time, the spectral profile is nearly linear and without the ballistic deficit being only slightly reduced in terms of energy due to the RC time, which, in our case, is much longer than the sampling time, and its effect is negligible. Case (C) represents the high-energy X-ray photons with energy much larger than the threshold. The threshold is crossed early in the rising part of the curve, and collection after the sampling time occurs before saturation. This results in collection before the electron reaches the anode, and ballistic deficits appear. This results in a low-quality spectrum. This fact is well documented in Figure 13, where we show the charge collection efficiency of the X-ray photon absorbed near the cathode. The collected charge dependency on the excited charge is calculated using Equation (8). We marked three energy regions in Figure 13, showing significant deviations from the optimum curve. The first region, with energy less than about 16 keV, does not contribute to charge collection. The reason is the energy threshold of 16 keV, which is not overcome by these photons (this corresponds to the photon energy marked with (A) in Figure 13). The second region of 16–45 keV shows a nearly linear profile, which is only slightly reduced relative to the optimum. These photons are collected, showing correspondingly reduced collected charge (the collection of photons, marked with (B), from this region, is shown in Figure 12). The third region is for high-energy photons with an energy greater than 45 keV. These photons reveal incomplete collection resulting from the large ballistic deficit. The collection of photons from this region, marked with (C), is shown in Figure 12. The ideal full-charge collection is shown in Figure 13 with a straight red line starting from the initial coordinates.

To obtain a better spectrum with the same low bias of 300 V, the sampling time should be appropriately extended. We followed such a scheme and calculated the X-ray spectrum in the presented model, extending the sampling time to 32 ns. The spectrum calculated under such conditions is plotted in Figure 14. It is evident that the spectrum is much better in this case. The weak deviation from the original spectrum is mainly due to the neglected holes, and the finite lifetime of electrons that was used is 20 μs in these simulations.

## 5. Conclusions

In this work, we have studied the polarization phenomena in the CdZnTe radiation detector induced by high-flux X-ray excitation. The electric field warping measured using the Pockels effect was modeled with a defect model solving coupled drift–diffusion and Poisson equation. We have developed a procedure to find minimal bias with optimal counting performance, which is still acceptable for spectroscopic applications, allowing for simpler electronic circuits and the easier portability of detector devices. We have defined the minimum bias of 300 V as still being acceptable for the application of a 2-mm-thick detector in X-ray spectroscopy at high-flux 20 Mcps/mm^2^ excitation and analyzed its performance. We have developed a numerical model that predicts counting and spectroscopic behavior in a polarized detector. We have shown that the warping of the X-ray spectrum comes from the delayed transit time caused by detector polarization compared to the sampling time of the counting electronics. We have proven such a feature by calculating the spectrum with an extended sampling time of 32 ns. The decrease in measured counts at low bias is not mainly caused by carrier trapping but by the ballistic deficit of low-energy X-ray photons, which do not cross the threshold level, frequent pileups, and charge sharing between pixels accentuated at the low electric field near the pixelated anode. The presented model may be used as a simple diagnostic tool in sensor characterization.

## Figures and Tables

**Figure 1 sensors-23-05681-f001:**
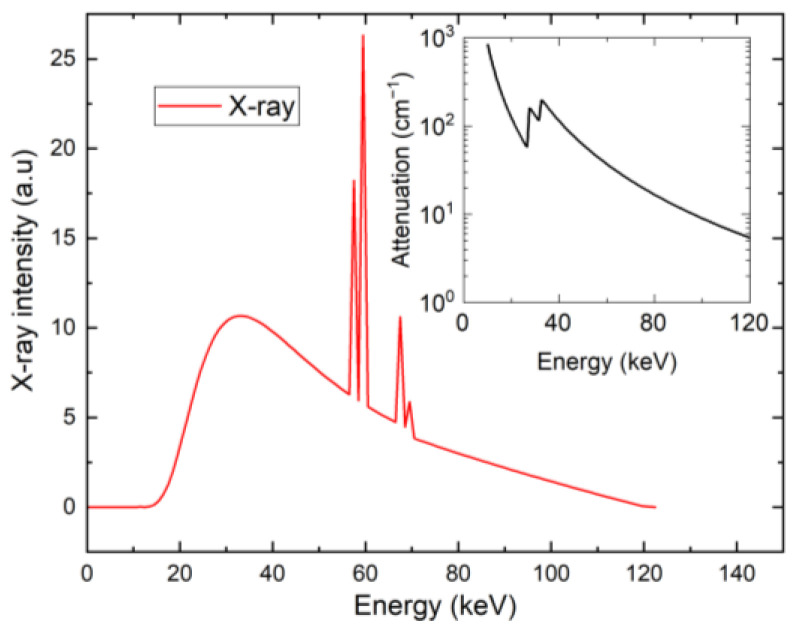
X-ray spectrum (red) with attenuation coefficient in CdZnTe (inset).

**Figure 2 sensors-23-05681-f002:**
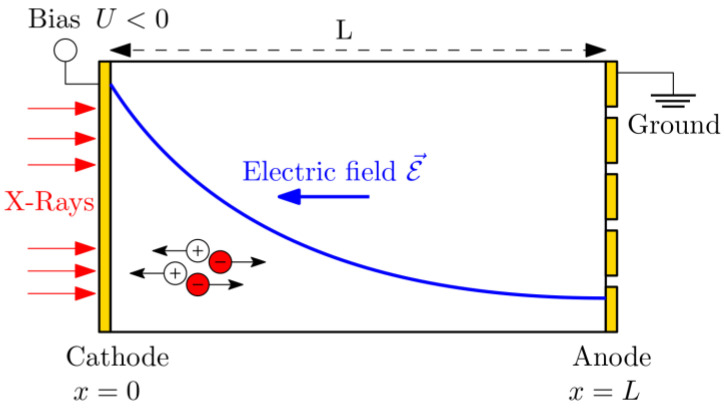
Schematic view of the electric field distribution in a detector with a positive space charge.

**Figure 3 sensors-23-05681-f003:**
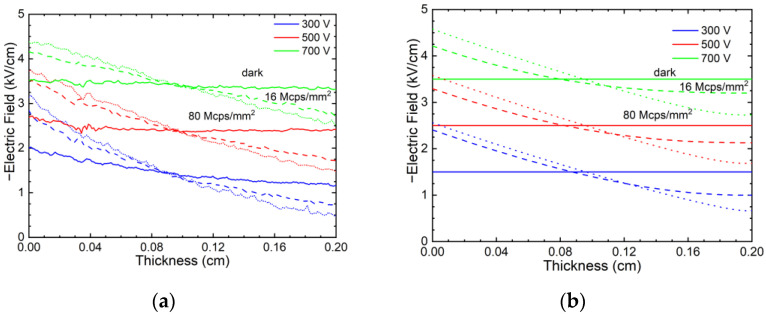
Measured electric field profiles in a 2-mm-thick CZT sensor (**a**), numerical simulation (**b**); solid lines represent electric field with no incoming X-ray (dark mode), dashed lines are for X-ray 16 Mcps/mm^2^, dotted lines are for X-ray 80 Mcps/mm^2^, blue, red, and green color mark 300 V, 500 V, and 700 V bias, respectively.

**Figure 4 sensors-23-05681-f004:**
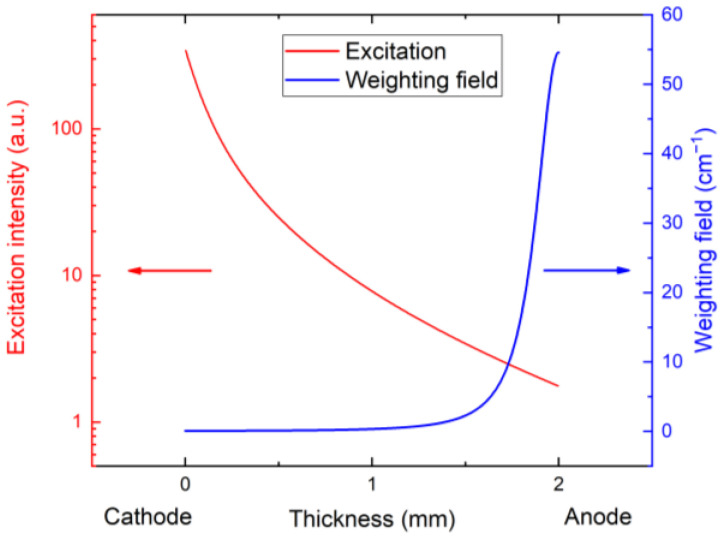
Profile of X-ray excitation, which shows dominant excitation under cathode with fast decrease toward anode. Weighting field was calculated according [30] in the middle of the pixel with 330 µm pitch.

**Figure 5 sensors-23-05681-f005:**
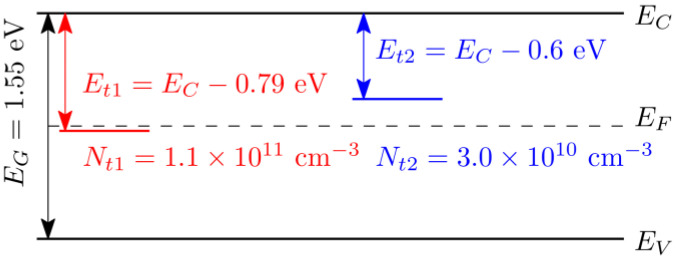
Scheme of energy levels with parameters determined by the fit of electric field. Hole trap is red; electron trap is blue.

**Figure 6 sensors-23-05681-f006:**
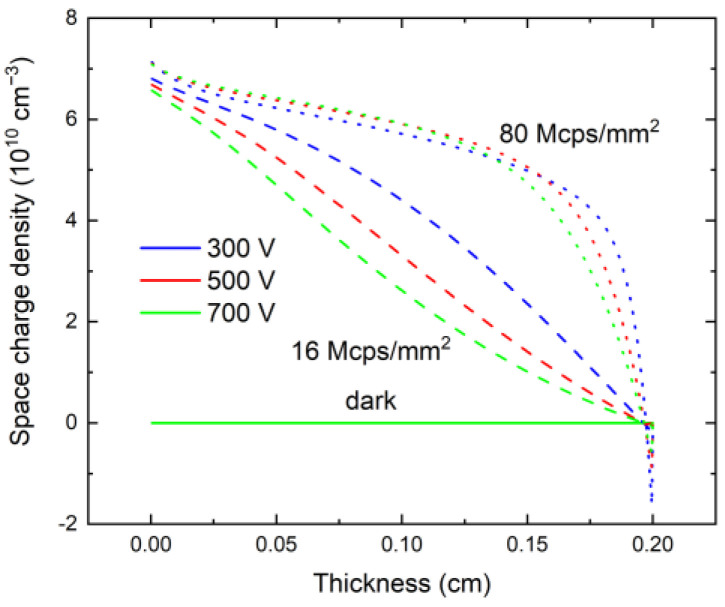
Calculated space charge density from the numerical simulation. In the dark regime, there is no space charge in the sample.

**Figure 7 sensors-23-05681-f007:**
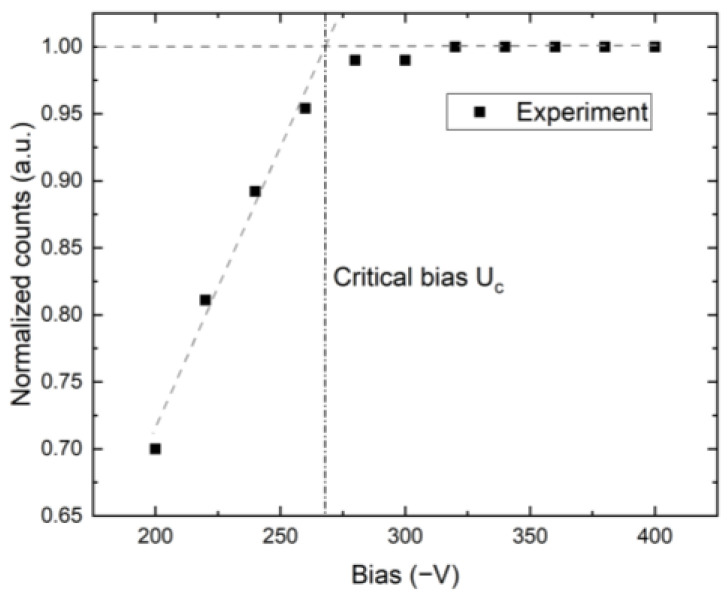
Measured counts in 2-mm-thick CZT sensor under X-ray conditions with a count rate of 20 Mcps/mm^2^ for a typical pixel. Procedure to extract critical bias $U_{c}$ is shown at the intersection of linear fit for bias $U<U_{c}$ with the horizontal line at maximum CCE.

**Figure 8 sensors-23-05681-f008:**
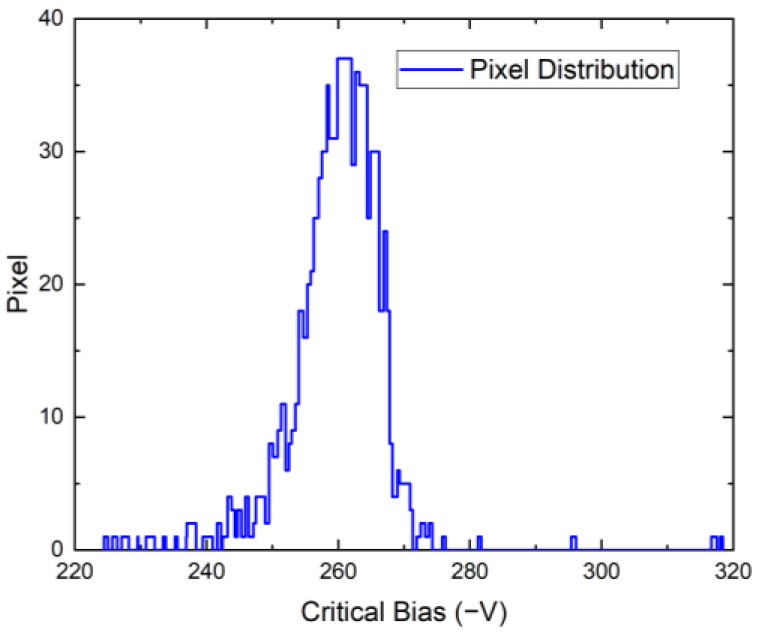
Distribution of $U_{c}$ values for all pixels.

**Figure 9 sensors-23-05681-f009:**
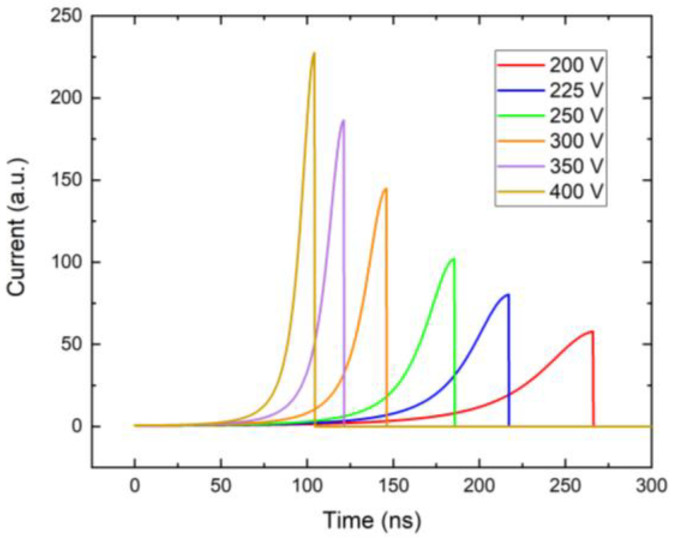
Simulated current waveforms for different biases and 20 Mcps/mm^2^ X-ray.

**Figure 10 sensors-23-05681-f010:**
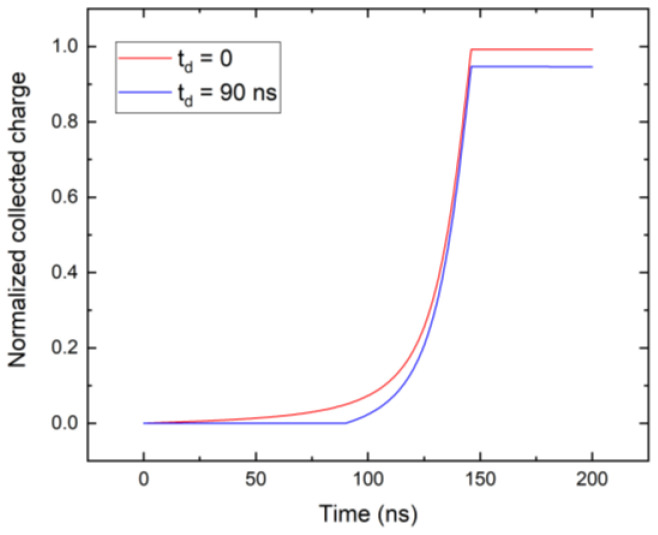
Comparison of the collected charge of X-ray photon absorbed near the cathode (red), and deep inside the detector, 1.44 mm (blue). U = 300 V.

**Figure 11 sensors-23-05681-f011:**
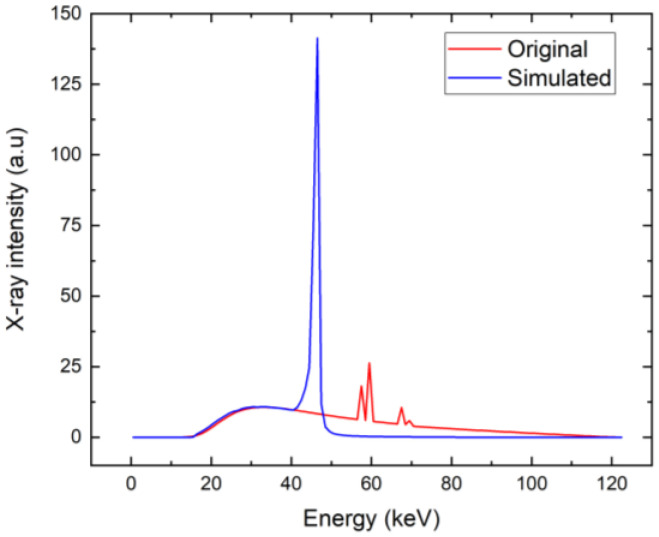
X-ray spectrum simulated for common parameters characterizing the electronic circuit, $t_{s}=16\;\mathrm{ns}$ (blue). The original X-ray spectrum is plotted for comparison (red).

**Figure 12 sensors-23-05681-f012:**
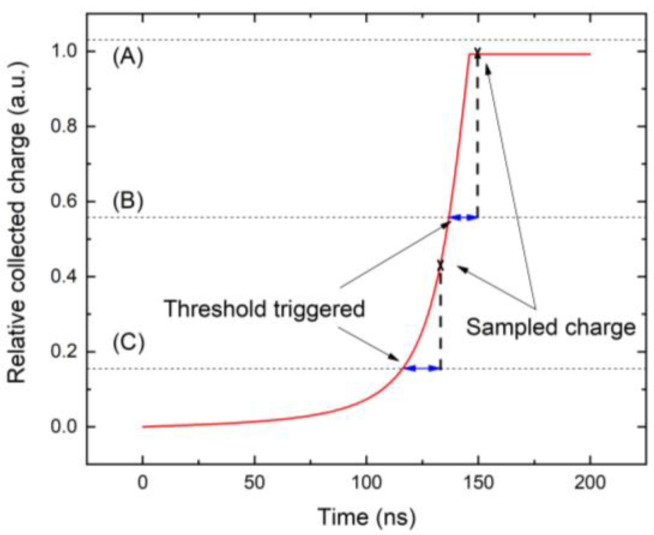
Analysis of sampled charge depending on photon energy. In case (A), the photon energy is lower than the threshold and the charge is not detected. In case (B,C), the threshold is triggered and the charge is sampled.

**Figure 13 sensors-23-05681-f013:**
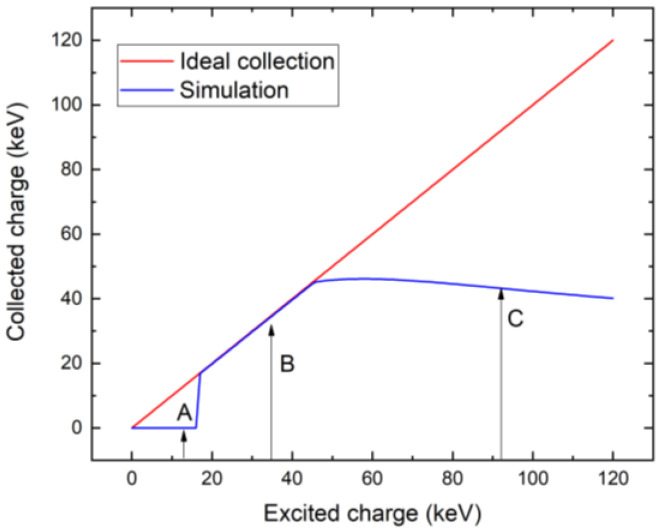
Dependence of the collected charge on the excited charge by X-ray photon absorbed near the cathode (blue) compared with the ideal full collection plotted with a straight line (red). Labels point to the threshold levels shown in Figure 12.

**Figure 14 sensors-23-05681-f014:**
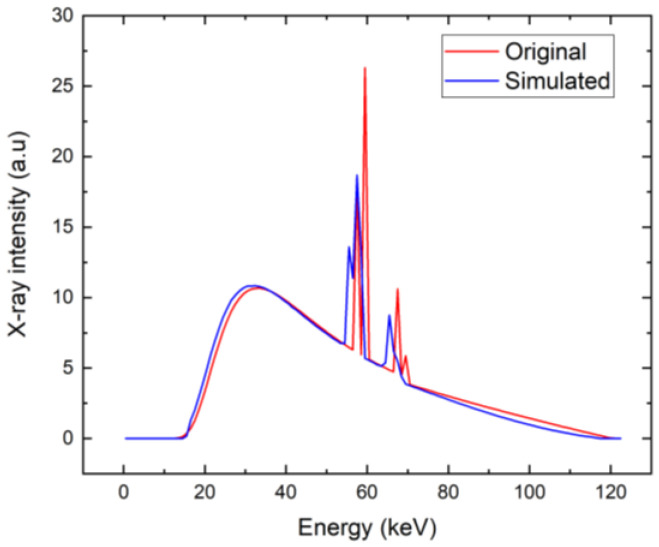
X-ray spectrum simulated for extended $t_{s}=32\;\mathrm{ns}$ (blue). The original X-ray spectrum is plotted for comparison (red).

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
