# Peer review of "Modelling Polarization Effects in a CdZnTe Sensor at Low Bias"

_sensors, 2023, doi:10.3390/s23125681_

Round 1
Reviewer 1 Report
This work studied the behavior of charge transport and collection under low electric field for CdZnTe sensors. Overall, the experiment is well designed, and the paper can be published in Sensors. Below are some comments.
1. Why do you choose CdZnTe as the sensing material?
2. The authors are encouraged to fabricate real devices to confirm their simulation results.
Reviewer 2 Report
This paper investigates the possibility of using a detector that can operate effectively under high-flux X-ray irradiation while maintaining a low electric field. The researchers conducted numerical simulations to visualize the electric field profiles affected by high-flux polarization. They developed a consistent defect model and simulated the charge transport in a commercial pixelated CdZnTe detector. The collected charge was evaluated, and the researchers analyzed the effect of allied electronics on the quality of the X-ray spectrum. They also suggested setup optimization for improving the spectrum shape.
Comment:
Introduction-
1- The first paragraph mentions the improvements in crystal growth and fabrication quality but does not specify the nature of these improvements. Adding a sentence or two to briefly explain these advancements would enhance the understanding of the reader.
2- The last sentence of the first paragraph is quite broad, mentioning the study of charge transport properties and pixelated detectors without specifying the specific findings or relevance to the current discussion. Providing a concise summary or highlighting the significance of these studies would make the paragraph more informative.
3- The last sentence of the second paragraph could be rephrased for clarity. Specifically, the phrase "an incomplete collected charge and count number are detected" could be modified to better convey the idea that reducing the electric field leads to a decrease in the amount of collected charge and the count numbers obtained.
Experiment-
4- The paper states that a semi-insulating commercial grade CdZnTe detector with dimensions 10 × 10 × 2 mm3 and a 330 µm pitch was studied. What were the reasons for selecting this particular detector? Were there any specific advantages or unique characteristics associated with this commercial-grade CdZnTe detector that made it suitable for the study?
5- It would be helpful to provide some more details about the characterization techniques used for the experimental measurements. Could you elaborate on the specific methods employed to measure the high-flux sensors and obtain the electric field profile? This information would provide a better understanding of the experimental approach.
6- What are the key metrics or parameters in evaluating the detector performance being assessed?
7- Could you explain how this ballistic deficit affects the accuracy and reliability of the detector's energy measurement? Are there any mitigation strategies in place?
Simulation of the detector performance at low bias-
8- Could you provide more information about the space charge effects being modeled in this paper?
Discussion-
9- Could you provide more information on why the authors disregarded electric-contacts-induced polarization? What are the implications of this decision on the accuracy of the simulations?
10- The electron mobility (µe) and hole mobility (µh) values mentioned are stated to be consistent with common values measured in CdZnTe. Could you provide any insights into how these mobility values affect the overall performance of the detector?
11- It is interesting to observe the distinction between the low-energy and high-energy regions in the spectrum. The peak at around 45 keV indicates a significant impact of the ballistic deficit on the charge collection. Are there any potential strategies or modifications that could be explored to mitigate or reduce the ballistic deficit and improve the collection efficiency for high-energy photons?
12- Are there any limitations or considerations to be aware of when implementing Ohmic contacts in real-world detector designs?
Overall, with some minor improvements in sentence structure, grammar, and organization, the text can effectively convey its intended message to readers.
